# The Role of Social Isolation and the Development of Depression. A Comparison of the Widowed and Married Oldest Old in Germany

**DOI:** 10.3390/ijerph18136986

**Published:** 2021-06-29

**Authors:** Franziska Förster, Melanie Luppa, Alexander Pabst, Kathrin Heser, Luca Kleineidam, Angela Fuchs, Michael Pentzek, Hanna Kaduszkiewicz, Carolin van der Leeden, André Hajek, Hans-Helmut König, Anke Oey, Birgitt Wiese, Edelgard Mösch, Dagmar Weeg, Siegfried Weyerer, Jochen Werle, Wolfgang Maier, Martin Scherer, Michael Wagner, Steffi G. Riedel-Heller

**Affiliations:** 1Institute of Social Medicine, Occupational Health and Public Health (ISAP), University of Leipzig, 04103 Leipzig, Germany; melanie.luppa@medizin.uni-leipzig.de (M.L.); Alexander.Pabst@medizin.uni-leipzig.de (A.P.); Steffi.Riedel-Heller@medizin.uni-leipzig.de (S.G.R.-H.); 2Department for Neurodegenerative Diseases and Geriatric Psychiatry, University Hospital Bonn, 53127 Bonn, Germany; Kathrin.Heser@ukbonn.de (K.H.); Luca.Kleineidam@ukbonn.de (L.K.); Wolfgang.maier@dzne.de (W.M.); Michael.Wagner@dzne.de (M.W.); 3German Center for Neurodegenerative Diseases, DZNE, 53127 Bonn, Germany; 4Medical Faculty, Institute of General Practice, Heinrich-Heine-University Düsseldorf, 40225 Düsseldorf, Germany; Angela.Fuchs@med.uni-duesseldorf.de (A.F.); Pentzek@med.uni-duesseldorf.de (M.P.); 5Medical Faculty, Institute of General Practice, Kiel University, 24105 Kiel, Germany; Hanna.Kaduszkiewicz@uksh.de; 6Center for Psychosocial Medicine, Department of Primary Medical Care, University Medical Center Hamburg-Eppendorf, 20246 Hamburg, Germany; c.van-der-leeden@uke.de (C.v.d.L.); m.scherer@uke.de (M.S.); 7Hamburg Center for Health Economics, Department of Health Economics and Health Services Research, University Medical Center Hamburg-Eppendorf, 20246 Hamburg, Germany; a.hajek@uke.de (A.H.); h.koenig@uke.de (H.-H.K.); 8Work Group Medical Statistics and IT-Infrastructure, Institute for General Practice, Hannover Medical School, 30625 Hannover, Germany; oey.anke@mh-hannover.de (A.O.); wiese.birgitt@mh-hannover.de (B.W.); 9Department of Psychiatry, Klinikum Rechts der Isar, Technical University of Munich, 81675 Munich, Germany; edelgard.moesch@tum.de (E.M.); dagmar.weeg@tum.de (D.W.); 10Central Institute of Mental Health, Medical Faculty, Mannheim/Heidelberg University, 68159 Mannheim, Germany; siegfried.weyerer@zi-mannheim.de (S.W.); jochen.werle@zi-mannheim.de (J.W.)

**Keywords:** widowhood, depressive symptoms, social isolation, old age, sex differences

## Abstract

Widowhood is common in old age, can be accompanied by serious health consequences and is often linked to substantial changes in social network. Little is known about the impact of social isolation on the development of depressive symptoms over time taking widowhood into account. We provide results from the follow-up 5 to follow-up 9 from the longitudinal study AgeCoDe and its follow-up study AgeQualiDe. Depression was measured with GDS-15 and social isolation was assessed using the Lubben Social Network Scale (LSNS-6). The group was aligned of married and widowed people in old age and education through entropy balancing. Linear mixed models were used to examine the frequency of occurrence of depressive symptoms for widowed and married elderly people depending on the risk of social isolation. Our study shows that widowhood alone does not lead to an increased occurrence of depressive symptoms. However, “widowed oldest old”, who are also at risk of social isolation, have significantly more depressive symptoms than those without risk. In the group of “married oldest old”, women have significantly more depressive symptoms than men, but isolated and non-isolated do not differ. Especially for people who have lost a spouse, the social network changes significantly and increases the risk for social isolation. This represents a risk factor for the occurrence of depressive symptoms.

## 1. Introduction

Depression is one of the most common psychiatric conditions in late life [1,2]. A systematic review and meta-analysis of the elderly reported high prevalence rates of 17% (95% CI 9.7%–26.1%) regarding dimensional measures of depression and of 7.2% (95% CI 4.4%–10.6%) regarding categorical measures [3]. The factors most closely associated with depression in old age are chronic illness, poor social networks and bereavement [4]. 

Bereavement, especially widowhood, is common in old age and can be accompanied by serious health consequences [5,6]. Previous studies have emphasized the impact of widowhood on depression in old age [2,7,8,9,10]. King et al. [10] showed that both widowed women and men aged 60 years and older developed significantly more depressive symptoms compared to their married counterparts. However, men might be more vulnerable compared to women [11]. Lee and DeMaris [11] show that widowed persons have lower levels of psychological well-being than those never married. Moreover, widowhood is often linked to substantial changes in social network and social support [12], and, associated with this, is also a risk factor for social isolation [13]. The definition of social isolation is subject to debate [14]. The present work understands social isolation according to Gironda and Lubben [15] as “an insufficient social network from which a person may draw from or exchange social supports”. Various studies showed a robust association between social isolation and worse health including chronic disease, cognitive decline [16] and depression [17]. These effects can be found for all age groups. Nevertheless, due to physical fragility and a certain need for assistance, the oldest-old are considered to be one of the most vulnerable groups in the population. Previous research has focused primarily on the younger aged people up to a maximum of 75 years [18,19], therefore the aim of the present work is to provide results for the group of the oldest-old.

Since widowhood occurs most often with increasing age, the oldest old represent an age group that deserves special attention. The United Nations’ World Population Aging report defines the oldest old population as those aged 80 years and over [20]. Social isolation is most likely, because other losses due to hospitalization, moving away to nursing homes or the death of significant others besides the partner, may additionally affect the integrity of the social network. Despite deserving special attention, there is an undersupply of oldest old individuals with psychosocial and mental health services [21,22].

There are consistent findings for clear sex-specific differences in both depression and widowhood. Research consistently shows that females are twice as likely than males to develop depressive symptoms [23]. Furthermore, due to their higher life expectancy, women in old age are more likely to be widowed than men. Additionally, women have also a lower probability of being subsequently remarried and have a greater risk of living alone in old age [11,24]. Studies also suggest that men and women react differently to widowhood [25]. Because of these differences, it is important that research also focus on sex differences in the individual factors.

So far, studies looking into the association between social isolation, widowhood and the development of depression symptoms in the oldest old are lacking. We identified only one paper: Golden et al. [19] explained a higher prevalence of depression in widows with the higher prevalence of loneliness and isolation. However, the median age of those investigated was 73 years. Studies usually do not cover enough individuals in their 80th and 90th, although oldest old comprise the growing population segment due to demographic change. Deeper knowledge of these relationships could help to identify specific at-risk groups among the oldest old who are at need for support and should be target for prevention services.

The present study aims to investigate the impact of social isolation on the development of depressive symptoms over time taking widowhood into account in a large German primary care sample of the oldest old. 

Therefore, the following research questions will be addressed:How high is the prevalence of social isolation in the widowed and married oldest old?Is social isolation associated with developing depression symptoms over time in widowed and married oldest old individuals?Are there other socio-demographic factors associated with developing depression symptoms over time in widowed and married oldest old individuals?

## 2. Materials and Methods

### 2.1. Study Design and Sample

In the current study, longitudinal data from oldest old individuals (80+) were used from the prospective “German Study on Ageing, Cognition and Dementia in Primary Care Patients” (AgeCoDe) and its extension “Study on Needs, Health Service Use, Costs and Health-related Quality of Life in a Large Sample of Oldest-old Primary Care Patients (85+)” (AgeQualiDe). Study recruitment started in 2003/2004 and includes an initial sample of *n*  =  3327. Detailed information on study design, inclusion and exclusion criteria have been described elsewhere [26,27]. 

The present study refers to longitudinal data from follow-up (FU) 5 to FU 9, as data regarding social isolation were only available in this period (time span: January 2011 to Nov 2016). Patients were included in the analyses, if they fulfilled the following criteria: (1) study participation at FU 5, (2) availability of complete data sets with regard to age, sex, education, family status (widowed or married), depression, and social isolation, (3) being non-depressed at FU 5 (geriatric depression scale-15 cut-off score of ≤ 5), (4) no cognitive impairments during the follow-ups (mini-mental state examination (MMSE) score ≥ 24), (5) no change in family status occurred between FU 5 and FU 9. 

Overall, the study sample consists of *n* = 679 oldest old individuals (Figure 1), which was further divided into two subgroups: “widowed oldest old” (*n* = 456) and “married oldest old” (*n* = 223) individuals.

### 2.2. Ethical Approval

The study was approved by the ethics committees of all six participating study centers and complies with the ethical standards of the Helsinki Declaration [28]. File reference numbers: Ethics Commission of the Medical Association Hamburg: OB/08/02 & 2817/2007; Ethics Commission of the University of Bonn: 050/02 & 174/02 for E 3.2 & 258/07; Medical Ethics Commission II, University of Heidelberg at the University Medical Center of Mannheim: 0226.4 & 2002 2007-253E-MA; Ethics Commission at the Medical Center of the University of Leipzig: 143/2002 & 309/2007; Ethics Commission of the Medical Faculty of the Heinrich-Heine-University Düsseldorf: 2079/2002 & 2999/2008; Ethics Committee of the TUM School of Medicine, Munich: 713/02 & 713/02 E). All participants provided written informed consent prior to their participation. 

### 2.3. Measurments

#### 2.3.1. Socio-Demographic Variables

We included sex (men and women), age and education as sociodemographic characteristics. Education was rated low, medium and high according to CASMIN (comparative analysis of social mobility in industrial nations) classification categories.

#### 2.3.2. Social Isolation 

Social isolation was assessed using the Lubben Social Network Scale (LSNS-6) [29]. The LSNS-6 was primarily developed for the older population and measures the size, closeness and frequency of contacts of a participant’s social network in terms of the perceived level of support from relatives and friends [30]. The total sum score of the six item version ranges from 0 to 30. Based on Lubben et al. [30], a cut-off score of ≤11 was used to define social isolation (yes vs. no). Only the information from the FU5 survey is included in the analyses, accordingly the variable does not vary in time.

#### 2.3.3. Depression

The GDS (Geriatric Depression Scale) is a suitable assessment instrument for depression in the general population of the oldest old [31]. The present study refers to the short version with 15 items (GDS-15) developed by Sheikh and Yesavage [32]. Participants are asked to respond with ‘yes’ or ‘no’ depending on how they felt in the past week. The GDS-15 sum score ranges from 0 to 15. Higher scores indicate more symptoms of depression. Pocklington et al. [33] conducted a systematic review and found a high sensitivity (0.89) and a high specificity (0.77).

### 2.4. Statistical Analysis

All analyses were conducted using Stata/SE 16.0 (StataCorp, LLC, College Station, TX, USA). Descriptive results are presented as absolute frequencies and percentage or mean ± SD, as appropriate. Differences between “widowed oldest old” and “married oldest old” individuals with regard to sociodemographic characteristics and social isolation were examined via Chi-Square tests (for nominal variables) or t-tests (for continuous variables). Mixed regression models were used to examine determinants of the trajectories for depression over time, both for “widowed oldest old” (*n* = 456) and “married oldest old” individuals separately (*n* = 223). Mixed regression models are especially suitable for analyzing panel data taking within- and between-subjects variance properly into account [34]. Both time-variant (age) and time-invariant predictors (sex, education, social isolation) were employed. In addition, the interaction between sex and social isolation was included in both mixed regression models.

Parish et al. [35] emphasize that entropy balancing is an effective method to eliminate covariate imbalances. Accordingly, this method was used to weight and match the socio-demographics (sex, age, education) of both “widowed oldest old” and “married oldest old” individuals before mixed regression models were applied.

## 3. Results

### 3.1. Patient Sample

In Table 1, the sociodemographic characteristics of the study sample are summarized. The mean age of the participants in the present study was 86.50 (SD 2.89) years, ranging from 82 to 98 years. Females consisted of 62.89 % of the sample (*n* = 427). The majority was low educated (56.41%). 

Inferential analyses revealed significant differences in age between “widowed oldest old” and “married oldest old” individuals (t = −5.4085, *p* < 0.001). Thus, the “widowed oldest old” were slightly older than “married oldest old”.

In addition, “married oldest old” were more likely to have more education than “widowed oldest old” (Chi^2^ = 13.6309, *p* < 0.001). Women were more likely to be widowed than men (Chi^2^ = 198.2190, *p* < 0.001).

### 3.2. Social Isolation in “Widowed Oldest Old” and “Married Oldest Old”

In Table 2, the prevalence of social isolation in the widowed and married oldest old are summarized. Of the “widowed oldest old”, nearly one-third was social isolated (30.70%). Whereas only one in five “married oldest old” lived in social isolation (19.73%). Thus, the two groups significantly differ from each other (Chi^2^ = 9.1242, *p* < 0.003).

### 3.3. Social Isolation and Depression Symptoms over Time

Table 3 presents the results of the mixed regression models for “widowed oldest old” and “married oldest old” individuals. For “widowed oldest old”, there is a significant difference in the frequency of depressive symptoms between widowed individuals with social isolation and those without social isolation (Coef. = −0.83; 95%-CI = 0.44; 1.23). There was no significant interaction effect between social network and sex.

“Married oldest old” with social isolation do not differ significantly in terms of frequency of depressive symptoms from those without social isolation. Additionally, there was no significant interaction between men and social isolation in this group.

### 3.4. Sociodemographic Factors and Depressive Symptoms over Time 

With regard to “widowed oldest old” individuals the following conditional effects could be identified: Higher age appeared to be a significant risk factor for the development of depression symptoms. In addition, it turned out that higher education is a significantly protective factor with regard to the development of symptoms of depression. 

However, in the group of “widowed oldest old” there is no significant difference in the frequency of depressive symptoms between men and women (Coef. = −0.38; 95%-CI = −0.84; 0.07). 

With regard to “married oldest old” individuals higher age appeared to be a significant risk factor for the development of depression symptoms. Education proved to be not a significant factor in this group. However, men in the group of the “married oldest old” had a significantly lower risk for the development of depression symptoms than women (Coef. = −0.68; 95%-CI = −1.31; −0.04).

Figure 2 shows the “average marginal effects” of depressive symptoms in widowed women and men, depending on the level of social isolation. There are significant differences between widowed women as well as widowed men with and without social isolation regarding number of depressive symptoms. Widowed individuals with social isolation have significantly more depressive symptoms than widowed participants without social isolation. Widowed men and women did not differ significantly in the frequency of depressive symptoms.

In addition, Figure 2 shows the “average marginal effects” of married women and men. In the married group, there was no significant difference between those with social isolation and those without social isolation, both in men and in women.

## 4. Discussion

The aim of this study was to investigate the impact of social isolation on the development of depressive symptoms over time taking widowhood into account in a large German primary care sample of the oldest old comprising 679 individuals in their 80th and 90th.

Overall, we found that “widowed oldest old” with social isolation developed more subsequent depressive symptoms than those without social isolation. Meanwhile, “married oldest old” with social isolation and those without social isolation do not differ in frequency of depressive symptoms. 

We found significant differences in sociodemographic characteristics between “widowed oldest old” and “married oldest old”. These differences in sex, age and education were also reported in other studies [36,37], and are also known as significant risk factors for the development of depressive symptoms [38]. 

In line with previous findings, the level of depressive symptoms in widowed old-aged men and women seems to be similar [2,25,39,40]. Moreover, the study by Lee et al. [9] showed that widowed men and women do not differ in the frequency of depressive symptoms.

In contrast to our results, Lee et al. [9] showed in a comparison between married women and men, that men have fewer depressive symptoms than women. In the present study, in the group of “married oldest old” were no significant differences in the frequency of depressive symptoms. 

We also found that various sociodemographic characteristics are associated with more depressive symptoms. In line with Kittel-Schneider and Reif [41] older age was associated with more depressive symptoms in the group of “widowed oldest old” as well as in the group of “married oldest old”. In contrast with Stein et al. [42] we found a significant association between levels of education and increased depressive symptoms in the group of “widowed oldest old”.

Many studies focus either on a comparison between widowed and married people or on the influence of social isolation on the development of subsequent depression. In the present study, however, both factors (widowhood and social isolation) were examined. 

Many studies have shown that social isolation and limited social support has been consistently linked to depression and depressive symptoms [43,44]. In our investigation of the oldest old in Germany, we can only partially confirm these results. Our study shows that “widowed oldest old” individuals with social isolation have significantly more depressive symptoms than those without social isolation. In contrast, there are no significant differences in the group of “married oldest old” individual. This means that “married oldest old” who are considered social isolated do not have more or less depressive symptoms than those without social isolation. 

The literature usually compares widowed and married individuals and rarely digs deeper. We found that isolated widowed men and women show significantly more depressive symptoms than widowed persons without social isolation. This finding suggests that oldest old German widowed are a vulnerable group. In contrast, no difference can be established for married people. Durkheim [45] famously emphasized that marriage has a protective effect. Additional studies [46] have shown that the social networks of married people are larger. Nevertheless, our study shows that married couples, even those who have a very limited, almost isolated network, do not differ in terms of the frequency of depressive symptoms from those with a larger network. Accordingly, is it simply important to have a partner at home, regardless of whether there are other friends? Mechakra-Tahiri et al. [47] and Luppa et al. [48] found that marriage was significantly associated with fewer depressive symptoms. However, Santini et al. [49] show that negative partner interactions were more significantly associated with various mental disorders. The discrepancy theory of loneliness suggests that people with few contacts (therefore considered to be isolated) might not feel lonely if the quantity and quality of the contacts are consistent with what they want. In addition, people with many social relationships (accordingly, not considered isolated) can feel lonely if the quantity and quality of these relationships do not match their desires [50].

### Strengths and Limitation

This is the first study which addresses the impact of social isolation on subsequent depressive symptoms considering marital status for the oldest age group of individuals aged 80 years and older. An important strength of this study is the comparison of widowed and married after controlling for potential differences in socio-demographics. To our knowledge, there are no studies comparing both groups with the same socio-demographic characteristics in terms of age, education and sex, since the group of widows is naturally older, and contains more women, who are (at least in many historical cohorts born in the first half of the twentieth century) less educated. Exactly these socio-demographic factors are themselves influencing factors for an increased risk of depression regardless of the widowhood status. 

We also used longitudinal data and excluded all people with depression in the first wave of research. This approach allowed for evidence of the occurrence of depressive symptoms during widowhood and during marriage. 

Our study is not without limitations. First, information in depressive symptoms was self-reported by the oldest-old individuals using the GDS-15. Nevertheless, to be able to estimate the prevalence of common mental disorders in widowed people, studies that use screening scales are also important [2].

Second, we only used LSNS-6 in our analysis of social isolation. Although Lubben et al. [25] show a high level of internal consistency and good convergent validity, it does not take into account the perceived loneliness and the quality of the social relationships. Further research should consider both, isolation as the more objective measure and loneliness as the subjective evaluation, at the same time [51].

## 5. Conclusions

Depressive symptoms in late life are common and are associated with enormous personal, social and economic burdens; thereby comprising an important public health problem [52]. Working with very old patients, a comprehensive focus should be placed on social connections. The present study has shown that individuals who have lost a spouse are especially at risk, as social network changes can increase the risk of social isolation potentially leading to an increase in depressive symptoms. These results show again that there is an urgent need for prevention and intervention approaches to reduce the psychosocial consequences of widowhood in old age. Individuals who have lost a spouse are especially at risk, as social network changes can increase the risk of social isolation potentially leading to an increase in depressive symptoms. In turn, both depressive symptoms, as well as social isolation, negatively contribute to chronic somatic conditions and worsen outcomes. Thus, general practitioners, in particular, should keep an eye on their patients’ social networks especially in light of the death of a spouse to help identify potential risk factors early. Furthermore, there is an increasing need for low threshold network-building intervention to promote adaptation after spousal loss for vulnerable oldest old individuals. Further generations may benefit from web-based bereavement interventions [53], which are currently developed and trialed in older age groups [54].

## Figures and Tables

**Figure 1 ijerph-18-06986-f001:**
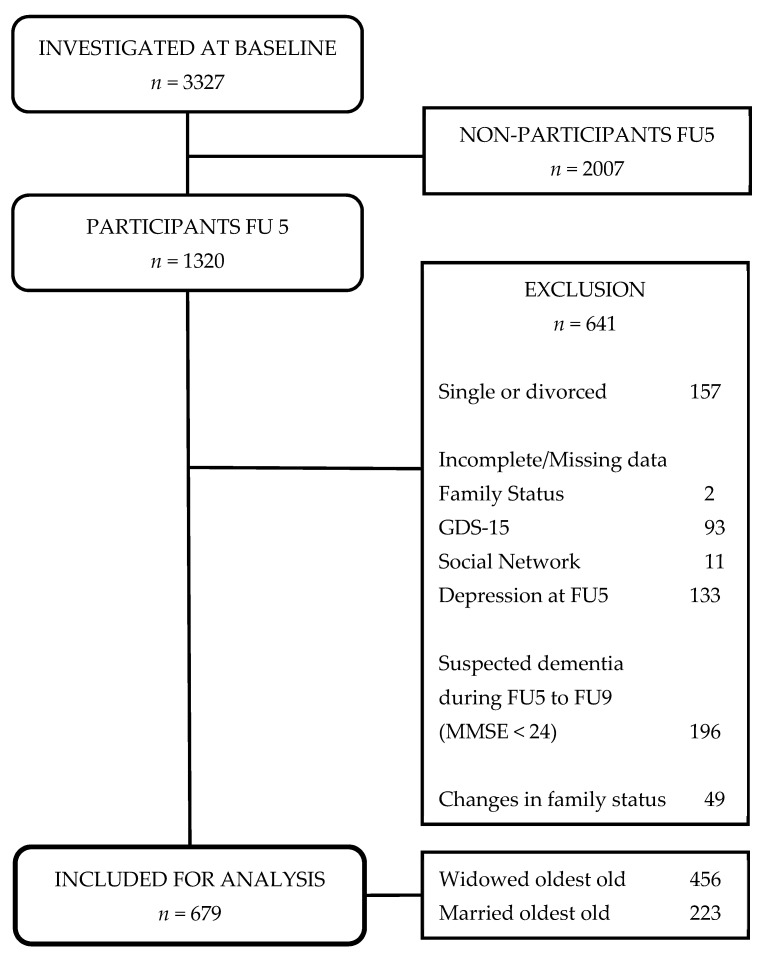
Sampling flowchart of the study. Notes: FU5 = follow-up 5, MMSE = mini-mental status examination.

**Figure 2 ijerph-18-06986-f002:**
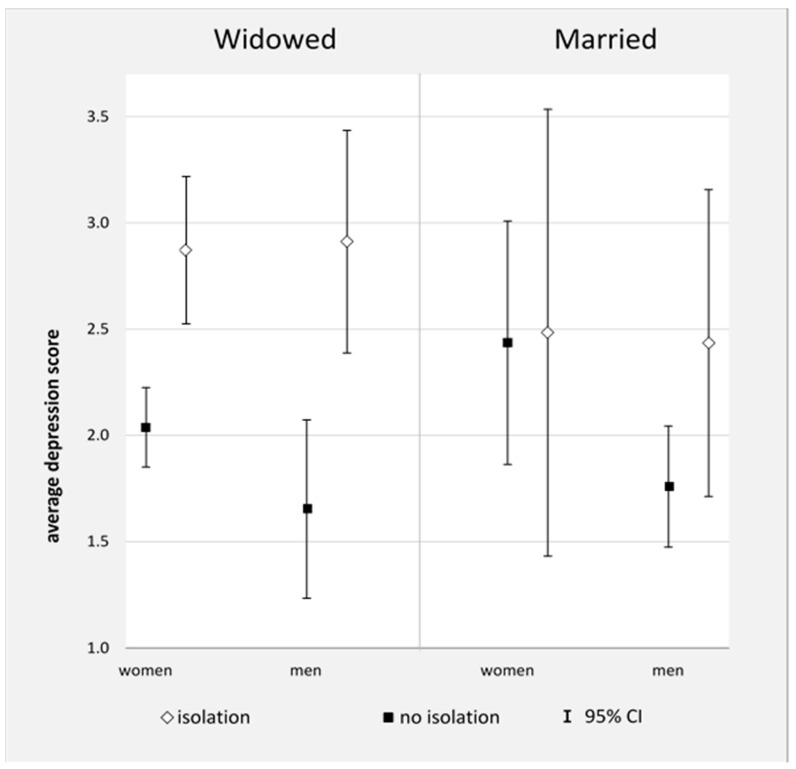
Average marginal effects of depression in the two subsamples “widowed oldest old” and “married oldest old” taking sex and social isolation into account. Notes: FU5 = follow-up 5, MMSE = mini-mental status examination.

**Table 1 ijerph-18-06986-t001:** Sociodemographic characteristics of the study sample (*n* = 679).

	FU5		Widowed Oldest Old		Married Oldest Old		*t*-Tests/Chi^2^-Tests
*n* (%)	679		456 (67.16)		223 (32.84)		
Age, mean (SD ^1^)	86.50	(2.90)	86.91	(2.97)	85.66	(2.51)	<0.001
Sex, *n* (%)							
Female	427	(62.89)	370	(81.14)	57	(25.56)	<0.001
Male	252	(37.11)	86	(18.86)	166	(74.44)	
Education, *n* (%)							
low	383	(56.41)	261	(57.24)	122	(54.71)	<0.001
middle	204	(30.04)	148	(32.46)	56	(25.11)	
high	92	(13.55)	47	(10.31)	45	(20.18)	

^1^ SD = Standard deviation, FU5 = Follow-up 5.

**Table 2 ijerph-18-06986-t002:** Social isolation in “widowed oldest old” and “married oldest old”.

	FU5		Widowed Oldest Old		Married Oldest Old		Chi^2^-Test
*n* (%)	679		456 (67.16)		223 (32.84)		
**Social isolation, *n* (%)**							
yes	184	(27.10)	140	(30.70)	44	(19.73)	0.003
no	495	(72.90)	316	(69.30)	179	(80.27)	

FU5 = Follow-up 5.

**Table 3 ijerph-18-06986-t003:** Predictors of depression symptoms * in the course of time in the two subsamples “widowed oldest old” and “married oldest old” ^1^.

	Widowed Oldest Old (*n* = 456)			Married Oldest Old (*n* = 223)		
	Coef. (95% CI)	*p*-Value	Wald	Coef. (95% CI)	*p*-Value	Wald
Sex (Ref.: women)						
men	−0.38 (−0.84; 0.07)	0.099	−1.65	**−0.68 (−1.31; −0.04)**	0.037	−2.09
Social isolation (Ref.: no)						
no	Ref.			Ref.		
yes	**0.83 (0.44; 1.23)**	<0.001	4.14	0.05 (−1.11; 1.21)	0.936	0.08
Interaction						
Men social isolation (yes)	0.42 (−0.35; 1.20)	0.283	1.07	0.63 (−0.75; 2.00)	0.370	0.90
Age *	**0.17 (0.13; 0.22)**	<0.001	7.98	**0.24 (0.12; 0.35)**	<0.001	4.06
Education	Chi^2^ = 5.80	0.055		Chi^2^ = 0.07	0.964	
low	Ref.			Ref.		
middle	**−0.38(−0.70; −0.07)**	0.018	−2.37	−0.08 (−0.65; 0.49)	0.788	−0.27
high	−0.24 (−0.69; 0.22)	0.312	−1.01	−0.03 (−0.63; 0.57)	0.922	−0.10

^1^ Results from two mixed regression models are presented. Both groups were balanced. Coef.: Coefficient; CI: Confidence Interval; Ref.: Reference Category; * age and depression (depend variable) are time varying variables, all the other variables are time-invariant. The bold type shows the reader at first glance which factors are significant.

## Data Availability

The data presented in this study are available on request from the corresponding author. The data are not publicly available due to privacy.

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
