# Peer review of "The Role of Social Isolation and the Development of Depression. A Comparison of the Widowed and Married Oldest Old in Germany"

_ijerph, 2021, doi:10.3390/ijerph18136986_

Round 1

Reviewer 1 Report

The idea is interesting to explore the relationship between depression and windowed or isolated situation among old people. But further improvements would be necessary.

  1. What is the oldest old?, the readers may be confused by this description.
  2. Depression is common among old people, particularly the very old people. Therefore it would be be more interesting to explore the social isolation and depression among younger population.
  3. Since the participants enrolled in this study were people aged more than 85+, who may be not so clearly to answer all the questionnaire. This may bias the results.

Reviewer 2 Report

This paper focused on the important topic for the older population. Many comments are listed as below:

  1. Introduction should provide more information regarding the topic, eg., the importance of social isolation for the older population, the reasons for focusing on the group of the oldest old, and the relationship between depression and widowhood.
  2. One  criterion for selecting the participants is no change in family status occurred between FU 5 and FU 9, I wonder if social isolation changed during the period of investigation? Is it appropriate to group the participants into isolation group and non0isolation group if so?
  3. If the classification of the participants into women and men groups is necessary or not when average marginal effects of depression in the two subsamples “widowed oldest old” and “married oldest old” taking sex and social isolation into account? If necessary, introduction about gendered comparison is insufficient.
  4. Discussion should be reinforced by adding other information, e.g., the implication of this study; the impact of sociodemographic factors on depressive symptoms

Overall, this study should a revision to some extent. 

Reviewer 3 Report

The paper entitled “The role of social isolation and the development of depression. A comparison of the widowed and married oldest old in Germany” describes a longitudinal study aimed to explore the impact of social isolation on the development of depressive symptoms in oldest old people.

I would like to congratulate the authors for their comprehensive work. The study is well-designed, structured and reported. The topic itself is of great interest to IJERPH and the study is methodologically sound. I only have some minor comments for the authors:

Study design and sample (lines 90-110). The authors achieve a remarkable sample of participants considering the profile aimed to study. However, it would be recommended to include a sample size calculation in order to provide an estimation of the confidence interval and statistical power of the final sample to address the three following research questions.

Ethical approval (lines 115-120). Could the authors provide the code number of their approved protocol?

Data Availability Statement (lines 366-370). I recommend the authors to delete the sample paragraph and state instead where or if the data are available.

Round 2

Reviewer 2 Report

Thank you for the detailed responses.  This version is acceptable.